# Precise Quantification of Molybdate In Vitro by the FRET-Based Nanosensor ‘MolyProbe’

**DOI:** 10.3390/molecules27123691

**Published:** 2022-06-08

**Authors:** Kevin D. Oliphant, Marius Karger, Yoichi Nakanishi, Ralf R. Mendel

**Affiliations:** 1Department of Plant Biology, Braunschweig University of Technology, 38106 Braunschweig, Germany; k.oliphant@tu-braunschweig.de (K.D.O.); marius.karger@tu-braunschweig.de (M.K.); 2Department of Applied Biosciences, Graduate School of Bioagricultural Sciences, Nagoya University, Nagoya 464-8601, Japan; nakanish@agr.nagoya-u.ac.jp

**Keywords:** MolyProbe, fluorescence resonance energy transfer FRET, molybdenum homeostasis, molybdenum metabolism

## Abstract

Molybdenum (Mo) is an essential trace element in all kingdoms of life. Mo is bioavailable as the oxyanion molybdate and gains biological activity in eukaryotes when bound to molybdopterin, forming the molybdenum cofactor. The imbalance of molybdate homeostasis results in growth deficiencies or toxic symptoms within plants, fungi and animals. Recently, fluorescence resonance energy transfer (FRET) methods have emerged, monitoring cellular and subcellular molybdate distribution dynamics using a genetically encoded molybdate-specific FRET nanosensor, named MolyProbe. Here, we show that the MolyProbe system is a fast and reliable in vitro assay for quantitative molybdate determination. We added a Strep-TagII affinity tag to the MolyProbe protein for quick and easy purification. This MolyProbe is highly stable, resistant to freezing and can be stored for several weeks at 4 °C. Furthermore, the molybdate sensitivity of the assay peaked at low nM levels. Additionally, The MolyProbe was applied in vitro for quantitative molybdate determination in cell extracts of the plant *Arabidopsis thaliana*, the fungus *Neurospora crassa* and the yeast *Saccharomyces cerevisiae*. Our results show the functionality of the *Arabidopsis thaliana* molybdate transporter MOT1.1 and indicate that FRET-based molybdate detection is an excellent tool for measuring bioavailable Mo.

## 1. Introduction

Molybdenum (Mo) is an essential trace element for animals, plants, fungi and most microorganisms [1]. Mo is catalytically inactive, but when it is bound to the scaffold molybdopterin (MPT), it becomes biologically active as the molybdenum cofactor (Moco). On one hand, Mo is required only in minute amounts, and the uptake of high concentrations of Mo results in toxicity. However, on the other hand, the depletion of Mo is lethal for most organisms [2]. Despite the Mo enzymes’ ubiquitous presence in all kingdoms of life, there are exceptions, such as the common yeast *Saccharomyces cerevisiae* that survive without the possession of Mo enzymes [3].

Mo is the lithosphere’s 54th most common element, with an average concentration of 1–2 mg/kg in soils [4] or 50 nM in terrestrial environments [5]. Though, local values reaching from 0.4 to 36 mg/kg have been reported. However, Mo is the 25th common element and the most abundant transition metal in the ocean, with a concentration of 10 μg/L [4] or 110 nM [5]. At the slightly alkaline pH (8.3) of the oxygen-rich sea, Mo is almost exclusively present in the biologically available oxyanion molybdate (MoO_4_^2−^) form, which is the predominant form at a pH higher than 4.2 [6]. Therefore, the state and bioavailability vary depending on the soil’s local pH. Soil Mo levels can severely increase due to industrial mining or agricultural activities [7,8].

On the other hand, cattle and sheep ingesting Mo-enriched plants show symptoms of molybdenosis, leading to the loss of hair and diarrhea [9,10]. The Institute of Medicine (USA) recommended a daily allowance of Mo intake for adult men and women of 45 μg/day in 2001. Studies have shown that the average dietary intake was 109 and 76 μg/day for men and women, respectively. The tolerable upper intake level was set to 2 mg/day based on impaired reproduction and growth observed in animals [11].

The plasma membrane of living cells forms an essential semipermeable barrier that allows the diffusion of small uncharged molecules, but larger molecules or ions cannot cross the membrane [12,13,14]. Therefore, the oxyanion molybdate needs to be transported in and out of cells via specific transmembrane transporters. In bacteria, molybdate transport is relatively well understood [15]. Eukaryotic molybdate transport is still poorly understood, and only recently, transporting systems in algae, humans and plants have been discovered [16,17,18]. In the plant *Arabidopsis thaliana*, the molybdate transporter type 1 (MOT1.1) mediates high-affinity molybdate transport (K_M_ ≈ 20 nM) and is localized in the plasma membrane, similar to the homologous protein LjMOT1 from the legume *Lotus japonicus* [18,19]. Thus, MOT1.1 is an essential molybdate importer from the soil and distributes the anion into the plant [19]. These proteins are within family one of the molybdate transporters (MOT1). The second family of molybdate transporters in eukaryotes (MOT2) belongs to the major facilitator superfamily (MFS) that includes uniporters, symporters, and antiporters [20]. However, similar to MOT1 family members, their transport mechanism is yet to be determined. Additionally, neither the role nor the localization of MOT2 transporters has been resolved yet [21].

One of us created a genetically encoded fluorescence resonance energy transfer (FRET) molybdate nanosensor named MolyProbe to investigate molybdate transporters within living cells. For this approach, fluorescence proteins were recombinantly fused to two copies of the molybdenum-binding domain (MoBD) of the molybdate-sensing transcription factor ModE from *Escherichia coli* [22]. Utilizing this tool, it was revealed that the human molybdate transporter HsMOT2/MFSD5, a homolog to the algal transporter CrMOT2 [23], has no direct influence on intercellular molybdate levels in vivo. Instead, it was observed that novel oxalate-sensitive and sulfate-resistant transporter(s) take up molybdate in a mammalian model cell culture [22]. In fungi, information regarding molybdate uptake and distribution is lacking, but a novel transporter belonging to the MOT1 family was annotated in silico in the fungus *Neurospora crassa* [16]. With respect to characterizing new transporters, one must quantify molybdate. Methods such as atomic emission spectrometry, inductively coupled plasma mass spectrometry (ICP-MS) and neutron activation analysis are very precise. However, they quantify only total Mo, not molybdate as the biologically active form in homogenized biological samples. Here, we present a way to use the MolyProbe nanosensor protein for in vitro tests. This assay is fast and reliable for quantifying molybdate content within solutions, media and cell extracts. Furthermore, we show the influence of molybdate supplementation on strains with and without these molybdate transporters by utilizing three different model organisms.

## 2. Results

### 2.1. Creating a Strep-Tagged MolyProbe

The previously created MolyProbe sequence contained an internal Strep-TagII sequence (WSHPQFEK) between the two MoBDs. We tried to purify the protein using this internal Strep-TagII, yielding no protein after affinity purification. As a result of this, the previous purification of the MolyProbe protein did not rely on affinity purification. Instead, a time-intensive four-step process was used [22]. To ease the purification, we created a new version of the MolyProbe using a terminal Strep-TagII sequence (Figure 1a). To predict the structure of the MolyProbe protein and to see if the N- or C-terminus was accessible for affinity purification, we used ColabFold (Figure 1b). The resulting structure prediction had a local distance difference test (pLDDT) score of >95%. The predicted MoBD sequences aligned well with the previously described structure for ModE, a protein involved in molybdate uptake from *E. coli*. The structure was solved in a complex with molybdate by using X-ray crystallography [24]. The fluorescence proteins also significantly aligned with GFP’s previously solved structure [25]. The structural prediction showed that both termini of the protein should be suitable for adding a recombinantly introduced Strep-TagII. We decided to add the Strep-TagII to the C-terminus of the protein, yielding MolyProbe-Strep.

### 2.2. Purification of MolyProbe-Strep

The MolyProbe-Strep clone was created exactly as described in the Materials and Methods section. After cloning, the plasmids were transformed into *Escherichia coli* NEB5α cells (New England Biolabs). A 30 h culture was used to express the protein and to purify it via Strep-TagII affinity chromatography for expression. The different fractions of the purification process were collected and separated in an SDS-PAGE (Appendix A), monitoring the success of the expression and the purity of the protein. The MolyProbe migrated at an estimated size of 85 kDa in the SDS PAGE. Extended purification with additional washing steps through high salt buffers in the affinity chromatography was used to optimize the purity of the protein yielded. As a next step, anion-exchange and size-exclusion chromatography were performed to characterize the purified protein and to perceive if the samples were free of other protein contaminations. The MolyProbe gene codes for two fluorescence proteins (FP), CFP and YFP, which have an absorbance peak at 434 nm (CFP) and 515 nm (YFP), respectively. These MolyProbe-specific absorbances could be monitored additionally at 280 nm. For anion-exchange column chromatography, one major peak with a MolyProbe-specific absorbance after ~30 mS/cm conductivity was reached (Figure 1c). The peak fronted slightly, pointing to an overloaded column or a possible sample degradation. The corresponding SDS-PAGE separated the peaks’ fractions (Figure 1d) and showed a prominent 85 kDa protein band, especially in the fractions C9–C11. Size-exclusion chromatography purification displayed a single major peak observable with slight tailing (Figure 1e). The SDS-PAGE of the fractions showed a similar picture compared to the anion-exchange chromatography protein (Figure 1f). Excitingly, fragmentation did not occur when the protein sample was not heated before loading the gel. Affinity purification yielded roughly 5 mg of protein per liter of expression culture.

### 2.3. Quantifying Molybdate Using an In Vitro FRET Assay

At first, it had to be assured that the Strep-tagged MolyProbe was working as intended. Based on the structural information of ModE, it was hypothesized that the domains change conformation significantly if molybdate binds to MoBD [24,26], which results in aligning the two FPs of MolyProbe-Strep, thus yielding increased FRET, whereas in the absence of molybdate, the FPs separate, causing reduced FRET (Figure 2a). As expected, adding 10 µM molybdate increased emission at 530 nm, whereas the emission at 475 nm decreased, changing the FRET ratio signal of the MolyProbe-Strep (Figure 2b). The response of the FRET ratio to molybdate covered a 1.9-fold range, which is remarkably high and should be sufficient for proper molybdate measurements, showing that the Strep-TagII does not influence the sensitivity of the assay. Our approach helped us design a 96-well-based assay with a high throughput of samples. The molybdate titrations for the calibration curves were created with MilliQ water in final concentrations ranging from 0 to 20 μM molybdate (Figure 2c). In particular, the derived FRET ratio grew in a sigmoidal matter with the added molybdate. For concentrations within 1 nM to 1 µM, the Hill equation fitted seamlessly with an R^2^ of 0.9974. The data points ranging in molybdate concentrations from 0.5 to 10 nM were neighboring tightly. Thus, precise molybdate determinations seemed difficult below 10 nM molybdate. Between 10 and 100 nM, the concentrations were significantly distinguishable. Since molybdate concentrations above 100 nM to 1 μM saturated the FRET signal, further measurements stopped at 100 nM, and higher concentrations needed to be diluted to fit within the measurement range. An amount of 5 to 50 µL of the unknown samples were diluted in three variations, 1:20, 1:40 and 1:100, to be within the detection range of the FRET assay. The reaction mix was filled up to a total volume of 50 µL with MilliQ water. Subsequently, 50 µL of the reaction buffer containing 60 nM MolyProbe-Strep was added, and the FRET signal was detected after 5 min (Figure 2d). With this assay established, we were able to vary and optimize the parameters of the assay.

### 2.4. Biochemical Parameters Influencing the FRET Assay

It was interesting to see how the MolyProbe behaved under various conditions for further characterization. Therefore, molybdate titrations with different MolyProbe quantities were carried out to optimize the sensitivity of MolyProbe depending on the protein concentrations. The MolyProbe was tested in concentrations ranging from 20 nM to 160 nM. Notably, the molybdate concentrations providing the half-maximal FRET signal (IC50) were altered MolyProbe-concentration dependent. Increasing the amount of MolyProbe lowered its sensitivity while increasing the maximum detection limit from 200 nM to 1 μM molybdate (Figure 3a).

For the daily use of MolyProbe, its long-term stability under different storage conditions is essential. The MolyProbe was stored for 4 weeks at different temperatures to gain insight into storage availability. The protein was kept on ice at 0–4 °C and was compared to shock freezing in liquid nitrogen with subsequent storage at −80 °C. These storage conditions were tested against freshly purified MolyProbe (Figure 3b). Both options resulted in no loss of effectivity. The lowest deviations were achieved with on-ice stored protein, but all storage options delivered satisfactory results.

Media and various cell extracts vary in their pH levels. We tried to alter only the buffering agents to see how pH changes the influence of MolyProbe activity. All substances stayed identical, except the MOPS-TRIS, which was substituted for either a 50 mM citrate buffer system (cit., pH 4.0 to 6.5), a 50 mM phosphate buffer system (phosp., pH 6.5 to 8.0) or a mixture of both buffering systems (cit.-phosp., pH 6.5). The pH-dependent results show that the measurement of molybdate via MolyProbe worked well in a pH range between 7.0 and 8.0 (Figure 3c). The titration curves at pH 6.5 show a significantly reduced FRET signal, and below pH 6.0, the FRET signal was declining. A FRET dynamic range of >1 was considered valid (Figure 3d). These results demonstrate a well-functioning molybdate measurement in buffers ranging from a pH of 6.5 to 8. The different buffer components had a minor but noticeable impact on the experiment, since the FRET ratio values were slightly shifted up with phosphate-based buffers compared to citrate buffers. Another observation was that citrate in the buffer lowered the sensitivity from 16 nM to 61 nM molybdate compared to phosphate. In addition, the MolyProbe’s emission spectrum at different pH and buffers showed that the FP emission was not present at a pH below 6 (Appendix A). Interestingly, the signal of CFP was impacted to a lesser degree. Based on these results, we proceeded to characterize the sensitivity of the in vitro FRET.

### 2.5. Determining MolyProbe’s Detection Range

The interpolation of the Hill equation resulted in more significant deviations than the interpolation of a linear detection range. Therefore, the linear range of the MolyProbe was determined using the reaction buffer at a pH of 8.0. The linear range was selected from 0 to 50 nM molybdate at a MolyProbe concentration of 20 nM (Figure 3e). Next, we wanted to determine the influence of a cell extract as a background signal. We prepared a dilution series of the 2 μM molybdate standard with crude yeast cell extract. Firstly, the molybdate standard was diluted 20×, 50×, 100× and 200× with yeast extract containing 20 mg/L protein to a final concentration of 8, 20, 40 and 100 nM, respectively. Next, the diluted samples were incubated with 50 µL reaction buffer containing 20 nM MolyProbe at a pH of 8.0 and were measured in a 96-well plate. The results were interpolated against the linear calibration curve and were finally extrapolated according to their appropriate dilution factors (Figure 3f). The measurements show that as the dilution factor increases, the measurement becomes more inaccurate. Diluting to 50 nM resulted in the lowest SD, and measuring at 10 nM resulted in more significant deviations, making the quantification less precise. These results show that a dilution within the higher end of the spectra should be considered to determine accurate molybdate concentrations. Our results demonstrate that precise molybdate quantifications are possible within crude extracts when the samples’ concentrations are in the MolyProbe’s linear range.

### 2.6. Measuring the Background of Molybdate in Water and Culture Media

It is known that even distilled water still contains Mo impurities. Therefore, water samples of different purities were measured via MolyProbe. The measurements were performed accordingly in quadruplicate. Double distilled water (_dd_H_2_O) had the lowest molybdate concentration with 13 ± 15 nM, followed by single distilled water (_d_H_2_O) with 21 ± 16 nM (Figure 4a). Tap water contained about 2.5-fold more molybdate traces, with a concentration of 52 ± 17 nM. The hydroponic culture medium (without additional molybdate) for the plant *Arabidopsis thaliana* contained slightly alleviated molybdate concentrations with 67 ± 21 nM. The culture medium supplemented with 100 nM molybdate had a concentration averaging 122 ± 21 nM. In summary, these results demonstrate that even double distilled water still contains molybdate impurities that must be considered when preparing “molybdate-free” media. These minute impurities can be easily detected via MolyProbe.

### 2.7. In Vitro Molybdate Quantification in the Plant Arabidopsis Thaliana

Earlier, MolyProbe was used to perform molybdate measurements in human HEK-293T cells in vivo [22]. However, organisms are complex biological systems, and their extracts differ from organism to organism and can potentially alter the FRET measurements via MolyProbe. For example, it was proposed that free molybdate is stored within the acidic vacuolar compartment of plants [27]. Therefore, we measured molybdate via MolyProbe in crude cell extracts of *Arabidopsis thaliana* wild-type strains cultivated in a hydroponic system with and without 100 nM molybdate. Additionally, a molybdate transporter (*mot1.1*) knock-out strain was cultivated under the same conditions. Previous results have shown that a knock-out of this transporter strongly reduces the Mo content within *A. thaliana* [18]. The green material of the plants was kindly provided by Jan Weber and colleagues (TU Braunschweig). The test was performed in triplicate with three to four biological replicates on three different days. The measurements showed the effects of a *mot1.1* knock-out (Figure 4b) on plants cultivated in a hydroponic system. Both strains without molybdate showed meager amounts of molybdate, an average of 107 ng per g fresh weight for the wild-type and roughly half with 57 ng per g fresh weight for the *mot1.1* knock-out. Additionally, the knock-out with supplemented molybdate showed a roughly 6-fold lower molybdate concentration within the cells than the wild-type. This experiment further confirmed the known function of MOT1.1 as a molybdate transporter and proved MolyProbe’s potential as a tool to tackle biological questions.

### 2.8. In Vitro Molybdate Quantification in the Yeast Saccharomyces Cerevisiae

Bakers’ yeast is a widely used model organism used to study the function of molybdate transporters. Especially the lack of Mo metabolism and MOT homologous genes makes the yeast an exciting model in Mo research [28]. The *Saccharomyces cerevisiae* strain YPH499 was grown with and without 230 μM supplemented molybdate (Figure 4c). Yeasts cultivated without molybdate showed a FRET signal with an average concentration of 53 ng molybdate per g fresh weight. On the other hand, samples cultivated with molybdate contained 503 ng molybdate per g fresh weight, roughly 10-fold more molybdate than the strain without molybdate. The FRET assay was performed in triplicate with three biological replicates. Overall, MolyProbe could differentiate between yeast grown with 230 μM molybdate and that grown without molybdate.

### 2.9. In Vitro Molybdate Quantification in the Fungus Neurospora Crassa

*Neurospora crassa* is a widely used model organism due to its relatively simple nutrient requirements, rapid growth, capacity to produce mutants and the ease with which it can be grown and cross-mated in culture [29]. Earlier in silico work has suggested that the NCU01356 gene codes for a hitherto unknown molybdate transporter [16]. Thus, for further investigation, to test MolyProbe’s capabilities, the NCU01356 knock-out was compared to the wild-type strain. Both strains were treated with 1 mM molybdate and without molybdate (Figure 4d). MolyProbe detected significantly reduced amounts of free molybdate in the fungi grown without adding molybdate. The knock-out did not have an impact on molybdate uptake. For 1 mM molybdate, both strains imported molybdate to a concentration of roughly 9.5 μg molybdate per g dry weight. Furthermore, the samples had to be strongly diluted with 1:160 and 1:250 to reach a measurable molybdate range and to reduce the interfering background signal (Appendix A). In contrast, NCU01356 did not impact molybdate uptake into the cell as its structural homolog MOT1.1 in *Arabidopsis thaliana* did, which is discussed below. The test was performed in triplicate with four biological replicates.

## 3. Discussion

Disturbances in cellular metal ion homeostasis are frequently associated with pathological alterations. In the case of the Mo oxyanion, excess amounts can potentially lead to molybdenosis, whereas the unavailability of Mo-dependent enzymes can be lethal [2,10]. Considering this circumstance, measurements of subcellular ion signals are of broad scientific interest. However, for this purpose, classical biochemical methods are complex, are often not feasible, or require large cell numbers. With the help of genetically encoded FRET probes, on the other hand, the visualization of metal ion dynamics within cells and their organelles is enabled [30].

The expression and purification of the MolyProbe with Strep-TagII are comparable to the earlier four-step purification [22]. The total yield was roughly 5 g per liter of expression, nearly half of what was achieved with the four-step purification (average of ~10 g/L, data not shown). However, since purification is significantly faster, affinity chromatography saves considerable time and effort.

FRET measurements require dedicated high-sensitivity devices capable of exciting fluorophores at a specific wavelength, while simultaneously detecting the distinct emission of multiple wavelengths. The MolyProbe’s dynamic range differed slightly on each run, and a fluctuating base level of molybdate in our experimental set-up or degrading MolyProbe may be the reason for these changes in the fluorescence ratio. Contrarily, the measurements of the different storage conditions seem to imply no significant sensor decay. However, a test of storage conditions was carried out with different batches of purified protein, not allowing a direct comparison between the sensor’s capabilities. Rather, qualitative observations of long-term storage impairments. Thus, time-dependent loss of sensor capabilities may still be possible.

Compared to previous publications, we tried to establish a method to quantify the total amount of free molybdate within cell extracts and solutions. We decided to try the in vitro quantification method within three different organisms, all of which have unique troubles regarding fluorescence spectroscopy. Analyzing *S. cerevisiae* led to the most reliable results directly after lysis. In the MolyProbe FRET assay, yeasts showed meager amounts of molybdate when grown without molybdate and significantly higher amounts when grown on molybdate. Since *S. cerevisiae* does not have a molybdate transporter, the authentic way of how molybdate is taken up by the cells remains open. However, it has been suggested that other anion transporters, such as the sulfate transporter, may be involved [17]. Additionally, the autofluorescence of riboflavin and other substances can interfere with the measurement [31]. When the samples were not diluted sufficiently, the autofluorescence became too intense, bleeding into the FRET assay.

Plant material from *A. thaliana* had a variety of autofluorescent substances, such as pigments, secretory compounds or structural components of cell walls. Therefore, it was feared that flavonoids and lignin would negatively impact the measurement data [32]. Autofluorescence was indeed present, but its interference was less pronounced than in the *S. cerevisiae* experiments. Using the MolyProbe assay helped us to demonstrate the relevance of the MOT1.1 transporter regarding Mo homeostasis in *A. thaliana*, as previously described [18,33]. Additionally, the molybdate concentrations in plants were apparently in an excellently measurable range. Interestingly, the measured difference in molybdate content between plants treated with and without 100 nM molybdate is consistent with the literature [10].

For the cultivation of *N. crassa*, we used a supplemented concentration of 1 mM molybdate, which was more than 4-fold higher than the concentration added to the yeast strains. Thus, a concentration of 1 mM is more than 1000-fold more elevated than the highest levels detectable with the MolyProbe, meaning that samples had to be diluted manifold. Additionally, for *N. crassa*, the autofluorescence interfered immensely with the measurements. This circumstance significantly affected measurements of cells with low molybdate amounts, as the YFP emission at 530 nm was additionally reduced. However, when sufficiently diluted, it was possible to show that the samples without additional molybdate had significantly less molybdate within the cells than those supplemented with it. Remarkably, the detected molybdate concentrations were 10-fold higher than in the *S. cerevisiae* experiments, although the amount of molybdate used was only 4-fold higher. As the knock-out of the NCU01356 gene, which codes for a putative molybdate transporter, did not affect molybdate concentrations measured in cell extracts, this result may mean that a so-far-unknown additional transporter should be located within the plasma membrane of *N. crassa*. Moreover, the protein encoded by NCU01356 is either not a molybdate transporter or may perhaps reside in the vacuolar membrane. The molybdate amounts within the *N. crassa* cells were roughly in the same range as the amount of Mo measured by ICP-MS (Appendix A Appendix A). The concentration detected by ICP-MS was always higher than the concentration determined with the MolyProbe assay. This may be because ICP-MS detects total Mo, whereas the MolyProbe assay detects free molybdate.

It was noted that the MolyProbe is generally very stable, as six months old samples stored at 4 °C are still fluorescent. Moreover, FRET assays with differently stored one-month-old protein showed no significant differences from newly purified MolyProbe. Thus, MolyProbe can be stored long-term at −80 °C, and in vitro assays can be performed after long periods. MolyProbe measurements were reliably possible between a pH of 7 to 8. At a pH of 6.5, the determination of molybdate concentrations was still possible, but the dynamic range decreased compared to higher pH values. Below a pH of 6, molybdate determination was not possible, and it was observed that the emission spectrum of the MolyProbe changed significantly. In particular, the YFP’s emission was altered substantially below a pH of 6, whereas the CFP’s signal remained stable until a pH of 5. These observations may be explainable due to pH-induced fluorescence quenching [34].

Overall, the data presented here demonstrate that the FRET MolyProbe assay is potentially extremely precise and very sensitive. Alternative modern detection methods such as atomic emission spectrometry, ICP-MS, or neutron activation analysis measure total Mo, whereas MolyProbe quantifies biologically active molybdate. Utilizing MolyProbe opens a fast and easy way to detect molybdate in vitro in cell extracts and media.

## 4. Materials and Methods

### 4.1. Strains and Materials

Na_2_MoO_4_ was bought from Sigma-Aldrich. All other chemicals and kits are described within the text. The model organisms and strains used in this study are listed in Table 1. The *S. cerevisiae* strain is YPH499 [35]. The *N. crassa* strains were purchased from the Fungal Genetics Stock Center (FGSC). The *A. thaliana* experiments were carried out with plants kindly provided by Jan Weber (TU Braunschweig, Braunschweig, Germany).

### 4.2. Genetical Construction of MolyProbe-Strep

The synthetic MolyProbe sequence previously created was used as a template [22]. The gene was amplified via PCR using F_c-strep.FOR (5’gaaccgcctttgtcagcagtgatgtaaacattgtgag3’) and F_c-strep.REV (5’attaactatgatggtgagcaagggcgag3’) as primers. Subsequently, the fragment was fused with a pTwoC vector containing a C-terminal Strep-TagII sequence [36], amplified using V_c-strep.FOR (5’ctgacaaaggcggttccgcctg3’) and V_c-strep.REV (5’ttgctcaccatcatagttaatttctcctctttaatgaattctgtgtgaaattg3’) as primers. Afterward, the MolyProbe-Strep construct was amplified via PCR using F_pBlue_strep.FOR (5’gaggtcgacttacttctcgaactgggggtggg3’) and F_pBlue_strep.REV (5’gaaactagtatggtgagcaagggcgag3’). Subsequently, the fragment was inserted into a pBluescript II SK (+) plasmid amplified via PCR using V_pBlue_strep.FOR (5’gagaagtaagtcgacctcgagggggg3’) and V_pBlue_strep.REV (5’tgctcaccatactagtttcctgtgt3’). All cloning was performed with the NEBuilder HiFi DNA Assembly Kit (New England Biolabs). The DNA assembly was carried out as described in the manufacturer’s manual, but the reaction volume was permanently reduced to a minimum. The plasmids were directly transformed into *E. coli* after finishing the DNA assembly.

### 4.3. E. coli Expression and Purification of Recombinant MolyProbe-Strep

*E. coli* cells were cultivated to express MolyProbe-Strep at 30 °C for 30 h without adding an inducer. The basal expression levels of the pBluescript plasmid levels were more than sufficient. Cold lysis buffer was added to the frozen cells with a total volume of ~30 mL. The cells were thawed gently in cold water. While thawing, complete proteinase inhibitor (Roche, Basel, Switzerland) and 2 μL of Benzonase (Jena Bioscience, Jena, Germany) per liter of expression culture were added. The cells were transferred into a homogenizer, EmusiFlex-C5 (Avestin, Ottawa, ON, Canada). The suspension was lysed thrice in the homogenizer with a pressure between 1.000 and 1.500 psi, keeping the lysate on ice. Additionally, the lysate was centrifuged for 45 min at 25,000× *g* and 4 °C. An amount of 2 mL of the StrepTactin Superflow high-capacity matrix (IBA) was added to a gravity flow column equilibrated with 2 column volumes (CV) of Buffer W (100 mM Tris-HCl pH 8.0, 150 mM NaCl). Then, the filtered (0.2 μm pores) cell lysate supernatant was loaded onto the column. Subsequently, washing was performed four times with 1 CV of Buffer W, three times with 1 CV of high salt Buffer W (100 mM Tris-HCl pH 8.0, 500 mM NaCl) and three times with 2 CV Buffer W. The protein bound to StrepTactin was eluted by adding 0.5 CV of Buffer E six times (Buffer W with 2.5 mM desthiobiotin). All purification steps were performed at 4 °C. After elution, the protein was concentrated by ultrafiltration using Vivaspinn Turbo 4 ultrafiltration units (Sartorius, Göttingen, Germany) with a molecular weight cut-off of 30 kDa.

### 4.4. Purification Analysis of MolyProbe-Strep

The concentrated protein sample was loaded onto an anion-exchange chromatography XK16-column with SOURCE 15Q resin (GE Healthcare, Chicago, IL, USA) for analysis. The CV was approx. 10 mL, and the column was operated at a flow rate of 1 mL per min and a pressure limit of 0.5 MPa. The sample was loaded and washed over 6 CVs. Elution was performed using a salt gradient from 50 mM to 1 M over 10 CV. Additionally, the protein sample was submitted to size-exclusion chromatography. The column used in this work was a HiLoad 16/600 Superdex 200 pg (GE Healthcare, Chicago, IL, USA). The column had a CV of 120 mL and was operated with a flowrate of ~0.7 mL/min.

### 4.5. MolyProbe In Vitro Assay

The MolyProbe assay was adapted from previous studies. The results show that the assay is capable of detecting molybdate while also being sensitive to tungstate; other anions were also detected but with a significantly lower sensitivity [22]. MolyProbe was excited at 430 nm, and the emission spectrum, 450–550 nm, was measured by a TecanSpark (Tecan, Männedorf, Switzerland). The emission intensity ratio (R_F530:F475_) was calculated from the fluorescent peaks of CFP (480 ± 5 nm) and YFP (530 ± 5 nm) using the Tecan software and exporting to Excel (Microsoft, Redmond, WA, USA). The data were fitted to a sigmoidal curve with the Prism software (GraphPad, San Diego, CA, USA).

For the assay, up to 50 µL of all samples, or 5 µL of the molybdate standard solutions, was pipetted into the wells adding MilliQ water up to 50 µL total. Afterward, 50 µL of the reaction buffer (20 mM MOPS-TRIS (pH8.0), 100 mM K-acetate, 2 mM MgCl_2_, 1 mM DTT and 0.025% (*w*/*v*) Tween-20 at 25 °C) containing either 40 or 60 nM MolyProbe was added to the well and was incubated for 5 min. The reaction buffer without MolyProbe was used as a blank for all measured samples.

### 4.6. pH Dependency of MolyProbe In Vitro Assay

The MOPS-TRIS buffer was exchanged for either a 50 mM citrate buffer system (cit., pH 4 to 6.5), a 50 mM phosphate buffer system (phosp., pH 6.5 to 8) or a mixture of both buffering systems (cit.-phosp., pH 6.5) while keeping all other components the same.

### 4.7. Neurospora Crassa Preparation

The *N. crassa* wild-type strain and the NCU01356 strain were cultivated according to [37]. Next, 10^6^ conidia was used to inoculate a 2 L flask and was subsequently incubated for 20 h at 30 °C and 140 rpm. Afterward, the biological material was harvested by filtering the flask’s contents through a Buchner funnel with filter paper (Miracloth, Merck Germany, Darmstadt, Germany). The cells were washed with 100 mL MilliQ water, and the cells were squeezed between paper towels to remove the medium. An amount of 1 g of semi-dry mycelium was then added to 50 mL Vogel’s Medium (VM) [38] with and without 1 mM molybdate (in 250 mL flasks). After another incubation for 5 h at 30 °C and 130 rpm, the cells were harvested as described above. The samples were transferred into 2 mL reaction tubes and were vacuum dried overnight to remove the remaining liquid. The samples were stored at −20 °C. An amount of 200 mg of mycelium was transferred into new 2 mL reaction tubes for the FRET assay, and 800 μL of ice-cold reaction buffer was added. Approximately 250 μL of 0.5 mm zirconia beads were added, and cells were lysed via the FastPrep-24 (MP Biomedical USA, Irvine, CA, USA) with the following parameters: 4 × 6.5 m/s, 30 s and 5 min breaks between. Afterward, the lysate was centrifuged at 16,000× *g* for 25 min, and the supernatant was transferred to a new reaction tube.

### 4.8. Saccharomyces Cerevisiae Preparation

The *S. cerevisiae* wild-type strain was inoculated overnight in a Yeast–Peptone–Glucose (YPD, 1% yeast extract, 2% peptone and 2% glucose) medium at 30 °C and 130 rpm. Next, the suspension was transferred to a new baffled flask with YPD medium and was diluted to an OD_600_ of 0.1. The cultures were either supplemented with or without 230 µM molybdate and were incubated for 5 h. Harvesting was performed by centrifugation at 4000× *g* for 5 min. The supernatant was removed, and the yeasts were washed twice with MilliQ water. After washing and removing the supernatants, the yeast cells were snap-frozen in liquid nitrogen and were stored at −80 °C. An amount of 200 mg of frozen yeasts were transferred into a new 2 mL reaction tube and were lysed, as described for *N. crassa* above.

### 4.9. Arabidopsis Thaliana Preparation

The *A. thaliana* samples were grown in a hydroponic system in Basal Nutrient Solution (BNS) medium [39]. When molybdate was added to the medium, it was supplemented with a concentration of 100 nM. Plants were harvested, their roots were removed and all other plant parts were homogenized via mortar and pestle, snap-freezing in liquid nitrogen and storing at −80 °C. The plant tissue was transferred into 2 mL reaction tubes for further preparation, and 700 μL of reaction buffer was added to the 100 mg sample. Approximately 200 μL 1.4 mm zirconia beads was added, and the plant tissue was lysed with the FastPrep-24 with the following parameters: 3× for 30 s at 6 m/s with 5 min breaks. After lysis, the sample was centrifuged at 16,000× *g* for at least 15 min. The supernatant was stored at −20 °C.

### 4.10. Data Analysis

After the FRET assay, the background signal or the autofluorescence of the biological samples was subtracted for all samples. Next, the ratio of the fluorescence peaks of the MolyProbe’s YFP and CFP was calculated:R_F530:F475_ = mean (F_530±5_)/mean (F_475±5_)(1)

A calibration curve was built with these values and the known molybdate concentrations of the standards. The data were fitted to sigmoidal Equation (2) via the GraphPad Prism software, as follows:
R_F530:F475_ = Bottom + ([MoO_4_^2−^]^HillSlope^) × (Top − Bottom)/([MoO_4_^2−^]^HillSlope^ + IC50^HillSlope^)(2)

The equation was used to interpolate the molybdate concentration. The initial concentration of the sample was calculated by extrapolating the dilution factors. Different dilutions of the sample were tested when the molybdate concentration was not in the measurable area (in the plateaus of the curve). The same was performed when autofluorescence was interfering. All measurements were performed at least as triplicates.

### 4.11. Structural Prediction Using ColabFold

ColabFold offers accelerated protein structure and complex predictions by combining the fast homology search of MMseqs2 with AlphaFold2 or RoseTTAFold [40]. The protein sequence of the predicted MolyProbe-Strep protein was used as a sequence query. For MSA, MMseqs2 (UniRef + Environmental) was used with an unpaired + paired approach. The model was run for 5 cycles, and the highest-ranked model (>85%) was used for the visualization of the structure in ChimeraX [41].

## Figures and Tables

**Figure 1 molecules-27-03691-f001:**
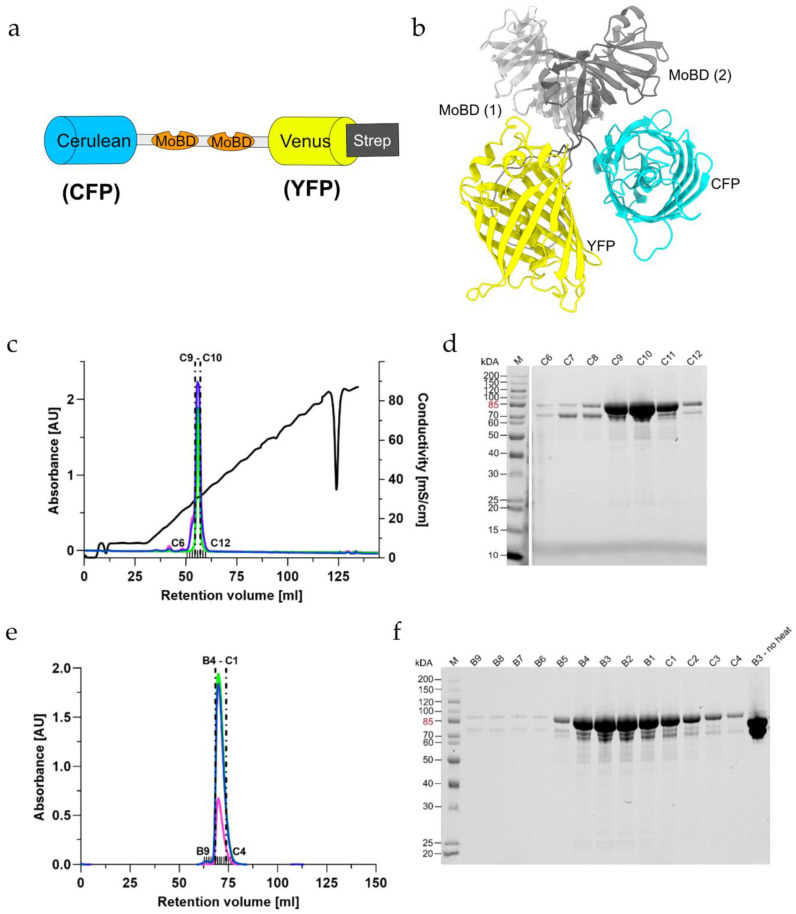
Purification of the MolyProbe-Strep protein: (**a**) Schematic description of the MolyProbe-Strep construct; (**b**) MolyProbe structure predicted with ColabFold; (**c**) Elution profile of the SOURCE 15Q (GE Healthcare) column; (**d**) SDS-PAGE of the selected elution fractions left to right: Ladder (M) (Pierce Unstained Molecular weight ladder, Thermo Fischer Scientific), C6 to C12; (**e**) Elution profile of a HiLoad 16/600 Superdex 200 pg (GE Healthcare) column; (**f**) SDS-PAGE of the selected fractions from left to right: Ladder (M), B9 to C4, fraction B3 not heated. Lines indicate absorbance in [AU] at (blue) 280 nm, (green) 434 nm, (pink) 515 nm and (black) conductivity in [mS/cm]. The analyzed fractions are marked above the *x*-axis, and the analyzed fractions are indicated with dotted lines.

**Figure 2 molecules-27-03691-f002:**
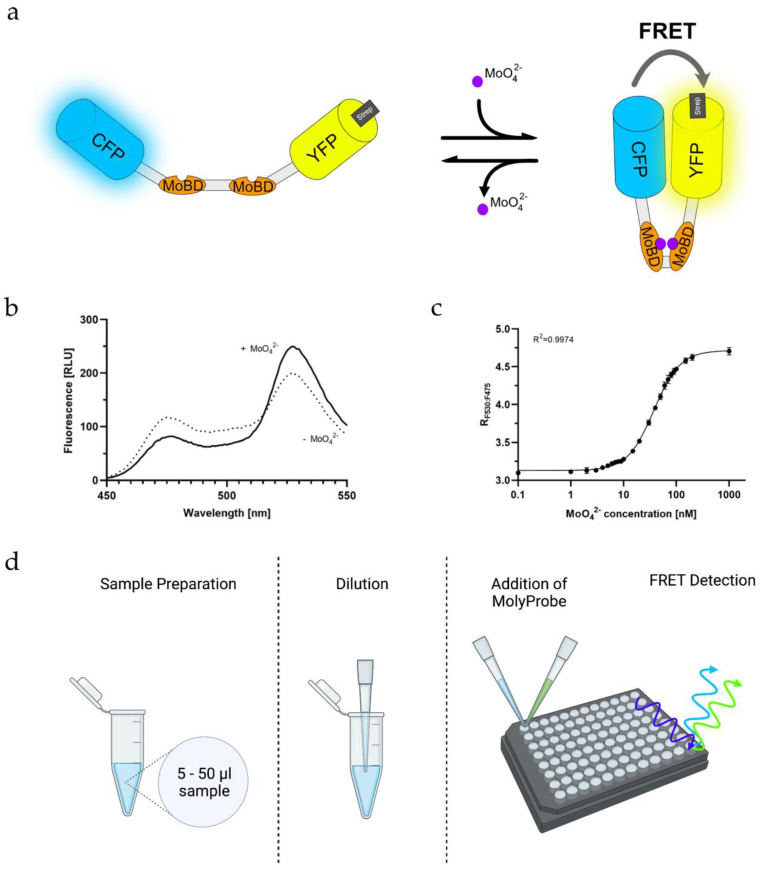
In-vitro-based molybdate detection assay: (**a**) Schematic representation of the MolyProbe-Strep reaction with molybdate, binding two molybdate ions per molecule; (**b**) Emission spectrum of the MolyProbe with (+) and without (−) 10 μM molybdate; (**c**) Large-scale measurement with 20 data points from 0.5 nM to 1 μM molybdate. The emission spectrum (450–550 nm) was measured at room temperature. An amount of 20 nM MolyProbe was used in reaction volumes of 500 μL (*n* = 3); (**d**) Preparation of a MolyProbe assay displayed, 5–50 µL sample was transferred into a new tube and diluted to 50 µL. This dilution step was repeated until an estimated molybdate concentration of 20–200 nM molybdate was reached, then 50 µL of the MolyProbe reaction buffer containing 40–320 nM MolyProbe-Strep was added to the mix. After 5 min, FRET was measured.

**Figure 3 molecules-27-03691-f003:**
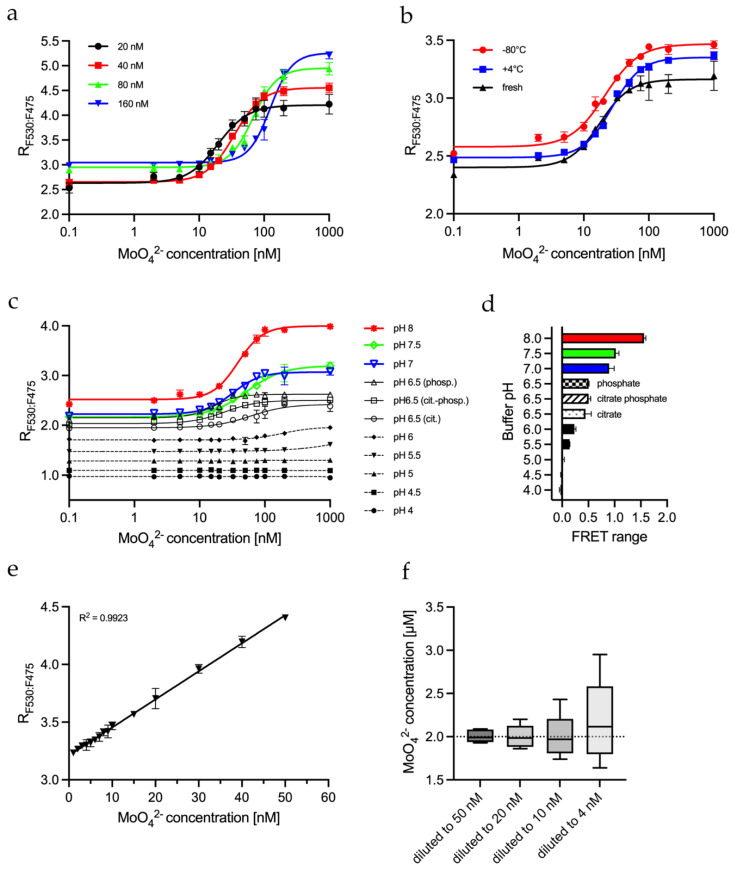
Characterization of MolyProbe-Strep parameters: (**a**) FRET ratio of a molybdate calibration curve analyzed with 20 nM, 40 nM, 80 nM and 160 nM MolyProbe; (**b**) Performance test of MolyProbe protein stored one month at +4 °C or one month at −80 °C, compared to freshly prepared protein; (**c**) FRET ratio was measured in different pH buffers (pH 4–6.5 citrate buffer, pH 6.5 citrate-phosphate buffer and pH 6.5–8 phosphate buffer); (**d**) The range of the FRET of the pH-dependent samples from (**c**); (**e**) Linear range of a concentration-dependent MolyProbe FRET assay; (**f**) Dilution-dependent accuracy of the MolyProbe FRET assay. An amount of 2 μM molybdate was diluted 40×, 100×, 200×, and 500× into the following concentrations, left to right: 50, 20, 10 and 4 nM, and they were extrapolated, respectively. Tukey boxes are plotted, showing the minimum, 25th percentile, median, 75th percentile, and maximum. An amount of 20 nM MolyProbe was used for all samples in a 100 µL reaction volume. The MolyProbes CFP (480 nm) and YFP (530 nm) peaks ±5 nm between 25–30 °C were measured. MolyProbe FRET ratio was plotted against the molybdate concentration. *n* = 3.

**Figure 4 molecules-27-03691-f004:**
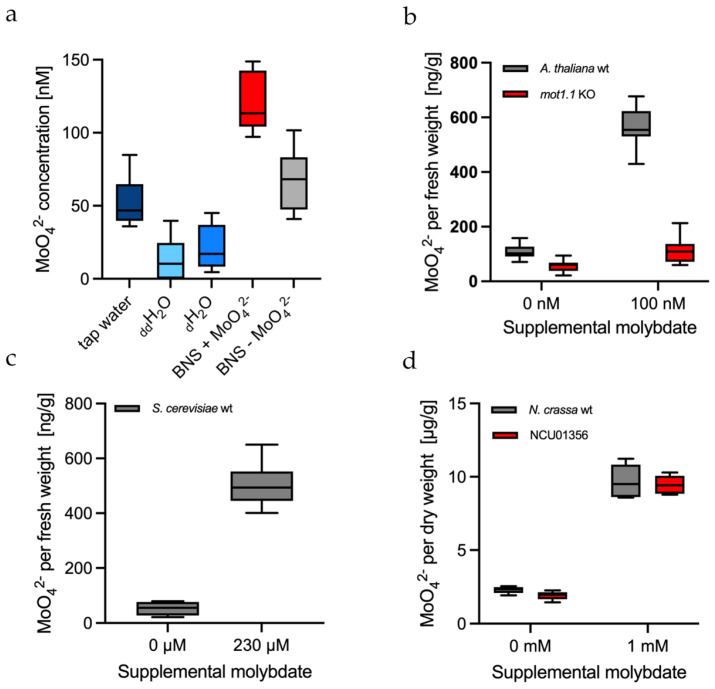
MolyProbe-Strep in vitro assay in culture media and cell extracts: (**a**) Water and culture media from left to right: tap water; _dd_H_2_O; _d_H_2_O; hydroponic BNS (see Materials and Methods) medium with 100 nM molybdate; hydroponic BNS medium without molybdate. *n* = 4; (**b**) Cell extracts of *Arabidopsis thaliana* plants cultivated in hydroponics with and without 100 nM molybdate. The roots were removed, and the remaining plant parts were collected, homogenized and measured in a 96-well plate. The *A. thaliana* strains that were used were molybdate transporter 1.1 knock-out (*mot1.1* KO) and the Col-1 wild-type (wt) strain. Biological replicates, *n* = 4 (without molybdate) and *n* = 3 (with molybdate); (**c**) Cell extracts of *Saccharomyces cerevisiae* strain YPH499 incubated with or without 230 μM molybdate. *n* = 3; (**d**) Cell extracts of *Neurospora crassa* strains incubated with or without 1 mM molybdate. *N. crassa* strains that were used were putative molybdate transporter knock-out NCU01356 and the wild-type strain 74-OR23-1V-A (wt). Biological replicates, *n* = 3. The Tukey boxplots show the minimum, 25th percentile, median, 75th percentile and maximum.

**Table 1 molecules-27-03691-t001:** Model organisms used in this work.

Organism	Strain Name	Genotype	Origin
*Saccharomyces cerevisiae*	YPH499	*ura3-52, lys2-801(amber), ade2-101(ochre), trp1-delta63, his3-delta200, leu2-delta1*	Sikorski and Hieter, 1989 [35]
*Neurospora crassa*	wild type NCU01356	*74-OR23-1V; mat A* *ncu01356::hph; mat a*	FGSC
*Arabidopsis thaliana*	wild type *mot1.1* KO	*Columbia-0 (Col-0)* *SALK_118311*	TU Braunschweig NASC

## Data Availability

Not applicable.

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
