# Peer review of "Precise Quantification of Molybdate In Vitro by the FRET-Based Nanosensor ‘MolyProbe’"

_molecules, 2022, doi:10.3390/molecules27123691_

Round 1
Reviewer 1 Report
This report presents an interesting and versatile improvement of a previously engineered molybdemum FRET probe. A few questions could be answered by the authors to improve the manuscript:
Line 68: From which organisms was the MolyProbe originally derived?
Line 86/Fig. 1: it would be easier for the reader to understand the improvements if the domain structure of the original Molyprobe was described or added to Fig.1. in this context: What is ModE (line 97)? Not all readers a Mo experts.
The careful characterization of sensitivity, range and outside factors modulating the probe repsonse are appreciated. However, it seems it is never properly be explained what exactly MolyProbe detects: really MoO42-? Or also Mo containing compunds such as molybdopterin or even MoCo? Would competition experiments with free and bound MoCo be possible? If it is exclusively MoO42-, then this should be cleary stated and referred to in the crude extract measurements as 'free' Mo.
It is assumed that the authors have considered this, but just to be sure: are the calibrations based on the two Mo molecules bound per probe as shown in Fig. 2?
The determination of Mo in media and crude extracts is indeed the major advantage provided by this manuscript. Could the absolute concentrations measured be compared with data from biochemical assays such as ICP-MS?
Author Response
Reviewer 1:
This report presents an interesting and versatile improvement of a previously engineered molybdemum FRET probe. A few questions could be answered by the authors to improve the manuscript:
Line 68: From which organisms was the MolyProbe originally derived?
We added the information to the text.
Line 86/Fig. 1: it would be easier for the reader to understand the improvements if the domain structure of the original Molyprobe was described or added to Fig.1. in this context: What is ModE (line 97)? Not all readers a Mo experts.
All tests were performed with the original MolyProbe as described in detail in the earlier paper of Nakanishi et al. (2013) including the ModE domains. We only added the Strep-tag. We reworded accordingly and explained ModE, the text clarifies the changes made, and we added an annotation for ModE.
The careful characterization of sensitivity, range and outside factors modulating the probe repsonse are appreciated. However, it seems it is never properly be explained what exactly MolyProbe detects: really MoO42-?Or also Mo containing compunds such as molybdopterin or even MoCo? Would competition experiments with free and bound MoCo be possible? If it is exclusively MoO42-, then this should be cleary stated and referred to in the crude extract measurements as 'free' Mo.
The MoBD domain was described in detail in the annotated papers. It was co-crystalized binding molybdate. Nakanishi tested the Probe against other anions. We added a sentence at lines 100 and 466. ‘Free’ was added where it seemed to be appropriated.
It is assumed that the authors have considered this, but just to be sure: are the calibrations based on the two Mo molecules bound per probe as shown in Fig. 2?
Yes, it has been considered. For calibration, specific amounts of molybdate where titrated against the FRET change observed in a photometer. It is not based on the stoichiometry of the reaction itself.
The determination of Mo in media and crude extracts is indeed the major advantage provided by this manuscript. Could the absolute concentrations measured be compared with data from biochemical assays such as ICP-MS?
We added a Supplementary Figure S4 with the ICP-MS MolyProbe comparison. Additionaly, we highlighted the differences that ICP-MS measured always slightly higher concentrations (line 388).
Reviewer 2 Report
This is a very interesting report describing the development of a fluorescent molybdate-binding protein as a tool for the quantification of molybdate in solution. The engineered protein “MolyProbe” contains two fluorescent domains (CFP and YFP domains) that are bridged by a molybdate-binding domain (MoBD). Upon binding of molybdate, MoBD forms a dimer and thereby brings the two fluorescent domains in close proximity to enable a FRET reaction. Unlike conventional elemental analysis that detects all Mo species indiscriminately, the MolyProbe-based method specifically detects molybdate. MolyProbe is shown to be sensitive for detecting molybdate at low nanomolar levels, and it displays a high stability in solution and during storage. The utility of the MolyProbe-based method in biological systems is demonstrated by in vitro assays in which the molybdate concentrations in the cell extracts of plant, yeast and fungus are determined, followed by verification by traditional ICP-MS analysis. Interestingly, these assays confirm the role of MOT1.1 as a molybdate transporter in A. thaliana as the molybdate level is greatly diminished in the mot1.1 KO strain.
The current study presents an innovative effort to develop a FRET-based method for the determination of cellular molybdate concentrations. The MolyProbe protein is cleverly designed and could inspire similar designs with different target binding domains that allow for the determination of other metals or chemical species in the biological systems. The technique is specific for molybdate rather than the total Mo species, and it represents a relatively easy and inexpensive alternative to standard elemental analysis, such as ICP-MS and ICP-OES. Perhaps even more excitingly, this technique could potentially be applied in vivo for the determination of molybdate concentrations of the cellular compartments across different biological systems. Because of these reasons, this manuscript warrants publication in Molecules.
There are some minor points for the authors’ consideration:
- It is not clear if metal ions similar to molybdate, such as vanadate or tungstate, may interfere with the detection of molybdate by MolyProbe. Can the authors provide any data on this or some insights into the potential non-specific reactions of MolyProbe with other metals?
- In the experiments involving biological systems (Figure 4), can the authors comment on the number derived from the MolyProbe-based method in comparison with those derived from other methods, such as ICP-MS?
- Can the authors expand the introduction to the molybdenum-binding domain (MoBD) and provide more references related to this topic? This background information is missing and its inclusion would greatly benefit the general readership.
Author Response
Reviewer 2:
There are some minor points for the authors’ consideration:
It is not clear if metal ions similar to molybdate, such as vanadate or tungstate, may interfere with the detection of molybdate by MolyProbe. Can the authors provide any data on this or some insights into the potential non- specific reactions of MolyProbe with other metals?
All these specificity tests have been described earlier in detail in the paper of Nakanishi et al. (2013). We added a sentence clarifying specificity (line4 66).
In the experiments involving biological systems (Figure 4), can the authors comment on the number derived from the MolyProbe-based method in comparison with those derived from other methods, such as ICP-MS?
We added a Supplementary Figure S4 with the ICP-MS MolyProbe comparison. Additionaly, we highlighted the differences that ICP-MS always had higher concentrations (line 388).
Can the authors expand the introduction to the molybdenum-binding domain (MoBD) and provide more references related to this topic? This background information is missing and its inclusion would greatly benefit the general readership.
We explained ModE in more detail, the text clarifies the changes made (line 100), and we added an annotation for ModE.